# Energy Management Strategy of Hybrid Ships Using Nonlinear Model Predictive Control via a Chaotic Grey Wolf Optimization Algorithm

Long Chen [ID], Diju Gao *[ID] and Qimeng Xue

Key Laboratory of Transport Industry of Marine Technology and Control Engineering, Shanghai Maritime University, Shanghai 201306, China; chenlong0019@stu.shmtu.edu.cn (L.C.); qmxue@shmtu.edu.cn (Q.X.)
* Correspondence: djgao@shmtu.edu.cn

**Abstract:** Reducing energy consumption and carbon emissions from ships is a major concern. The development of hybrid technologies offers a new direction for the rational distribution of energy. Therefore, this paper establishes a torque model for internal combustion engines and motors based on first principles and fitting the data collected from the test platform; in turn, it develops a model for fuel consumption and carbon emissions. Furthermore, the effect of irregular waves using an extended Kalman filter is estimated as well as feedback to the controller as a disturbance variable. Then, a parallel hybrid ship energy management strategy based on a new real-time nonlinear model of predictive control is designed to achieve energy conservation and emission decrease. A hybrid algorithm of chaotic optimization combined with grey wolf optimization is utilized to solve the nonlinear optimization problem in the nonlinear model predictive control strategy and a local refined search is performed using sequential quadratic programming. Through the comparison of fuel consumption, carbon emissions, real-time performance, and the engine load path, the superiority of the nonlinear model predictive control energy management strategy based on the chaotic grey wolf optimization algorithm is verified.

**Keywords:** sequential quadratic programming; nonlinear model predictive control; energy management; grey wolf optimization; extended Kalman filter

## 1. Introduction

According to the data from the International Maritime Organization (IMO), global $CO_2$ emissions from the shipping industry exceeded one billion tons in 2022, accounting for about 2% to 3% of total global emissions. As a result, the IMO has put forward higher requirements for energy saving and emission reduction in ships [1]. The emergence of hybrid ships offers a novel option for ship operators. As a result of the design of multiple power sources, hybrid power ships can maximize the dynamic performance of the ship through energy optimization and achieve the purpose of energy saving and emissions reduction. However, the multiple power sources and complex systems of hybrid ships create difficulties in ship energy management, where various constraints and additional degrees of freedom must be satisfied. Furthermore, the existence of various types of subsystems leads to the need for optimal control at different scales. It is a challenge to rationalize the ship's energy management in this state. Moreover, ship hybrid power systems are increasingly emphasized and supported by the manufacturing of the shipping industry and national policies [2]. Therefore, the research and implementation of ship energy optimization technology are imminent.

There are several approaches to hybrid ship energy management strategies available today, such as dynamic programming (DP), Pontryagin's minimum principle (PMP), the equivalent consumption minimization strategy (ECMS), and model predictive control (MPC) [3]. Certainly, current research on energy management strategies also includes

learning-based approaches. For example, various machine learning-based energy prediction methods are objectively analyzed in references [4,5] and the results achieved are analyzed. However, machine learning algorithms require extensive training and further verification is needed to determine whether their real-time performance can be guaranteed. However, MPC is a more effective method for research on energy management strategies because of its ability to simultaneously handle state and control multiple variables with apparent real-time and optimization effects. MPC can optimize the performance index by performing a series of control actions on the system in a predicted time horizon. For example, MPC strategies are used in the references [6,7] for energy management. At the same time, MPC is also used in combination with other methods for corresponding optimization. For example, a method combining the equivalent consumption minimization strategy (ECMS) with MPC was designed in the reference [8]. The ship's fuel consumption minimization and optimal power distribution are achieved. In reference [9], an adaptive MPC strategy is designed by combining recursive least squares (RLS) with MPC, which reduces the power loss of the ship by about 15% through online identification of RLS.

However, hybrid ships are typically a nonlinear system [10] which means that the classical MPC algorithms are not very good at achieving their objectives. Therefore, nonlinear model predictive control (NMPC) was proposed. It is similar to MPC in that they are both model-based and constraint-based optimization approaches and both achieve optimization by calculating control actions over a future time horizon. But it differs in that the NMPC prediction model and constraints can be nonlinear and contain time-varying parameters; relevant objective functions can be non-quadratic. For example, reference [11] proposed an NMPC-based energy management strategy for ships and exhibits its excellence.

Although the high complexity problem of hybrid ship systems can be solved by NMPC, it also suffers from low solution accuracy and low efficiency. Hence, it is important to consider how to maintain high solution accuracy and efficiency while using the NMPC for the energy management of ships. There are many methods available for solving NMPC problems. For example, the most classical sequential quadratic programming (SQP) is used to deal with the constrained optimization problems through the highly simulated Newton method. It approximates the Hessian matrix of the Lagrangian function using the quasi-Newton method during each major iteration, which in turn generates the quadratic programming (QP) subproblem, decomposing it in the search direction of the search process prior [12]. A combination of a simulated annealing algorithm and a fast feed-forward controller (QFFC) is proposed in reference [13] to solve NMPC at a given initial value relying on Monte Carlo methods. But the speed of solving NMPC optimization problems is not improved by this approach. The improvement in NMPC solution efficiency lies in how to achieve lower computational complexity during its iterative update. In reference [14], the optimization problem is defined as two-layer programming and decoupled from the dynamic nonlinear programming problem by using the optimality PMP condition to reduce dimensions and update states through MPC. The methods in the references [15,16] are similar in that they are based on ADMM solving and ALR solving, respectively, with the former requiring less computation time and the latter requiring fewer iterations and information exchange, both of which increase computational efficiency.

Heuristic algorithms were used by some scholars to improve MPC to enhance solving performance. In reference [17], a cooperative bat algorithm (CBA) is designed to be applied to the coordinated balancing NMPC problem of a network system, which showed powerful performance in comparison with the original BA algorithm and particle swarm optimization (PSO). Furthermore, reference [18] proposes a joint genetic algorithm and ant colony optimization (GA-ACO) for MPC, where dynamic trajectory tracking control is implemented by solving a standard MPC problem with constraints and combining it with dynamic sliding mode control (SMC). Therefore, new attempts and improvements to the NMPC solution to improve the stability of the NMPC energy management strategy in optimization and to alleviate the high demands on online computation is a crucial issue to be addressed.

The current use of NMPC to solve the ship energy optimal problem faces problems such as poor real-time control performance and distribution optimality [19]. As the classical SQP method relies heavily on selecting the initial value, when the initial value is not appropriately chosen, it will lead to longer optimization time, fall into local optimal solutions, or even fail to find feasible solutions. Thus, many researchers are making some improvements to the SQP problem. For example, an improved SQP is used in reference [20] to solve the MPC energy optimization problem; indeed, the computational efficiency and optimality of the SQP during the iterative process are improved. Improvements for SQP are also combined with Heuristic algorithms. In reference [21], the genetic algorithm is combined with the SQP algorithm to achieve a faster optimization speed than the single GA. The reference [22] similarly designed the GA joint SQP algorithm and verified the correctness of the method by solving three nonlinear dual singularity problems.

Heuristic algorithms are widely used to optimize industrial sites [23]. Given that Heuristic algorithms tend to converge faster than other algorithms, the computational effort can be reduced. Still, they also possess a certain degree of stability, which in turn guarantees the existence of optimal solutions. The grey wolf optimizer (GWO) has the characteristics of fast optimization, high accuracy, and strong robustness compared with the genetic algorithm and particle swarm algorithm. Some classical engineering class optimization problems can be solved by GWO [24]. In reference [25], an energy management method was designed using GWO to apply a hybrid energy source of supercapacitors and fuel cells. However, GWO also suffers from the problem of easily falling into local optimal solutions with slow convergence [26]. But chaos is characterized by randomness, convenience, and regularity [27]. Therefore, a chaotic algorithm is introduced into the population initialization and global search to improve the search efficiency of the algorithm and to make use of as much information in the solution space as possible. An L1 penalty term was added to prevent the global search from going beyond the boundaries in this study.

In summary, a novel parallel hybrid ship energy management strategy based on NMPC via a chaotic grey wolf optimization algorithm is proposed in this paper. Meanwhile, this study is also an extension of reference [11]. In reference [11], a trade-off parameterized energy management strategy based on NMPC is proposed and it also faces the condition of imaginary solutions when the initial values are not properly chosen. Therefore, this study innovatively combines classical optimization algorithms with heuristic algorithms to avoid this problem and applies it to ship energy management. In this work, firstly, an NMPC controller is designed based on the parallel hybrid ship dynamics model. The influence of irregular waves was also investigated using extended Kalman filtering. Then, a hybrid algorithm combining chaotic optimization and grey wolf optimization is used to find the optimal solution to the optimization problems, while the classical SQP algorithm is used for a local exact search in order to guarantee the accuracy of the solution. Finally, by comparing the solving algorithms commonly used in the nonlinear model predictive control, the performance of the designed algorithm is verified and the effectiveness of the proposed new nonlinear model predictive control energy management strategy is shown.

## 2. Hybrid Power System Description and Modeling

The actual hybrid power ship considered in this work is shown in Figure 1 and the powertrain of this hybrid power ship is shown in Figure 2, which features a typical parallel structure. The main parameters of the ship are listed in Table 1. The powertrain architecture of the hybrid power ship involves multiple subsystems and various factors of the ship's operation. So, the multi-input and multi-output (MIMO) model of a parallel hybrid power system is developed, which consists of the following power supply side components: (1) propulsion plant (mechanical shafts and the propeller); (2) internal combustion engine (ICE); (3) motor; and (4) battery. And the propeller provides a thrust at the power demand side. The ship's operation at the power demand side involves overcoming various types of resistance while maintaining speed.

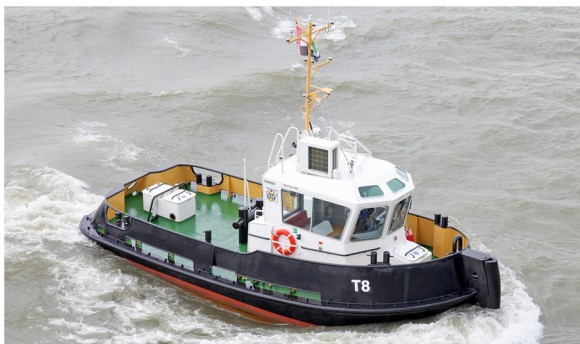

**Figure 1.** The tug ship considered in this work, model DAMEN Stun Tug 1205 from Damen Shipyards Group [28].

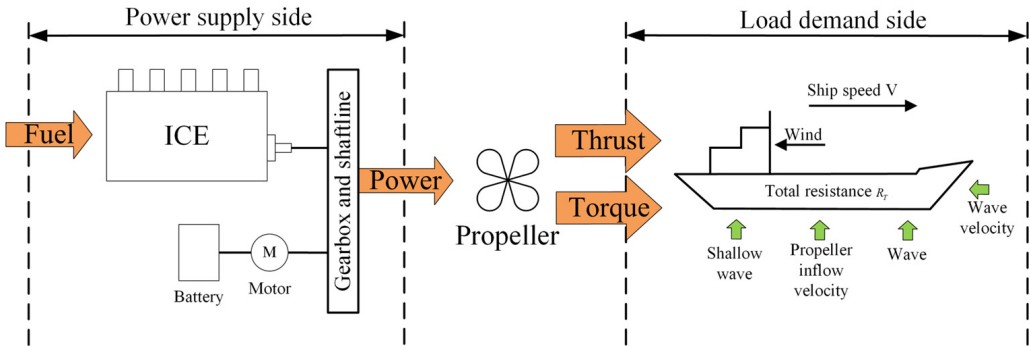

**Figure 2.** Powertrain architecture of a parallel hybrid power ship.

**Table 1.** Main parameters.

| Description | Symbol | Value |
|---|---|---|
| Ship length | $L_{ship}$ | 13.08 m |
| Ship breadth | $B_{ship}$ | 5.28 m |
| Draught aft | $H_{ship}$ | 1.85 m |
| Mass | $m$ | 55 t |
| Added-mass | $m_s$ | 2.75 t |
| Thrust deduction coefficient | $t_d$ | 0.145 |
| Effective wake coefficient | $f_d$ | 0.157 |
| Wetted area | $S$ | 341.5 m$^2$ |
| Gearbox reduction ratio | $\eta_r$ | 3.82:1 |
| Diameter of propeller | $D$ | 1.05 m |
| Air resistance coefficient | $C_{air}$ | 0.83 |
| Water resistance coefficients | $C_F + C_R$ | 0.0045 |
| Water density | $\rho$ | 1025 kg $\cdot$ m$^{-3}$ |

### 2.1. Ship Propulsion Plant Model

The modeling of a ship propulsion plant needs to take into account the power supply and load demand shown in Figure 2. And the model of the ship propulsion plant is derived from the reference [11].

### 2.1.1. Propeller Model

Based on the definition of four-quadrant open water characteristics [29], a propeller radius of 70% and a hydrodynamic pitch angle are used:

$$\beta = arctan\left(\frac{V_p}{0.7\pi Dn_P}\right) \tag{1}$$

where $n_P$ is the linear speed of the propeller. $V_p$ is the propeller inflow velocity subjected to the ship speed $V$ and the wave turbulence velocity $v_d$. So, $V_t$ is defined as:

$$V_t = (1 - \delta_d)V + v_d \tag{2}$$

where $\delta_d$ is the wake fraction.

Therefore, the torque $Q_P$ and thrust $T_P$ of the propeller are calculated as follows:

$$\begin{cases} Q_P = \frac{\pi}{8}\rho C_Q\left(\sqrt{V_t^2 + (0.7\pi n_P D)^2}\right)D^3 \\ T_P = \frac{\pi}{8}\rho C_P\left(\sqrt{V_t^2 + (0.7\pi n_P D)^2}\right)D^2 \end{cases} \tag{3}$$

where $C_Q$ and $C_P$ are the torque coefficient and thrust coefficient, respectively.

2.1.2. Irregular Wave Model

Modeling methods for irregular waves from reference [11]. According to the principle of superposition, irregular waves can be formed by the superposition of regular waves. Firstly, the wave height $\zeta(t)$ is defined as:

$$\zeta(t) = \sum_{i=1}^{\infty} A(\omega_i)cos(\omega_i t + \varpi_i) \tag{4}$$

where $i$ denotes the $i$-th wave and $A(\omega_i), \omega_i$ and $\varpi_i$ are the magnitude, the frequency, and the phase of the regular wave, respectively. The total number of waves is as follows: $\kappa_i = \frac{2\pi}{L}$ ($L$ is the length of each wave).

Then, suppose the wave encounter angle is $\Psi$. Thus, the excitation wave frequency $\omega_{v_i}$ is defined as:

$$\omega_{v_i} = \omega_i - \kappa_i V cos(\Psi) \tag{5}$$

So, the wave turbulence velocity $v_d$ is given as:

$$v_d(t) = -\sum_{i=1}^{\infty} \zeta(t)e^{-\sigma\kappa_i}\omega_{v_i}sin(\omega_{v_i}t + \varpi_i)cos(\Psi) \tag{6}$$

where $\sigma$ is the depth of immersion for the propeller hub.

2.1.3. Ship Hydrodynamics

The surge motion of a ship is defined as follows:

$$(m + m_s)\dot{V} = k_P T_P(1 - t_d) + R_T + F_w \tag{7}$$

where $k_P$ is the number of propellers and $m$ and $m_s$ are the ship's mass and the added mass on the ship, respectively. $t_d$ is the thrust deduction coefficient and $R_T$ is the total resistance, which contains the various resistances shown in Figure 1. $F_w$ is the resistance of waves.

2.2. *Modeling of Rotational Dynamics and Internal Combustion Engine*

The rotational dynamics of a power plant are derived from Equation (8)

$$\dot{\omega}_e = \frac{1}{J_s}(T_{ICE} + T_M - D_{load}) \tag{8}$$

where $J_s$ is the moment of inertia at the power supply side, $T_{ICE}$ is the brake torque of the internal combustion engine, $T_M$ is the output torque of motor, and $D_{load}$ is the disturbance torque load that is applied to the power supply side.

According to reference [30], the brake torque $T_{ICE}$ of the ICE and the engine fuel consumption ($\dot{m}_f$) can be modeled depending on the torque distribution ratio $r_{ice}$ of the ICE and engine speed $N_e$.

Therefore, $T_{ICE}$ can be expressed as follows:

$$T_{ICE} = a_i \begin{bmatrix} N_e^2 & N_e & r_{ice} & 1 \end{bmatrix}^T \tag{9}$$

where $a_i$ and $i$ are the coefficients of the approximate polynomial and the $i$-th coefficient, respectively.

The engine fuel consumption $\dot{m}_f$ is expressed as follow:

$$\dot{m}_f = b_j \rho_f \begin{bmatrix} N_e^2 & r_{ice}^2 & N_e r_{ice} & N_e & r_{ice} & 1 \end{bmatrix}^T \tag{10}$$

where $b_j$ and $j$ are the coefficients of the approximate polynomial and the $j$-th coefficient, respectively. $\rho_f$ is the density of diesel fuel.

The brake specific fuel consumption (BSFC) map [31] is displayed in Figure 3.

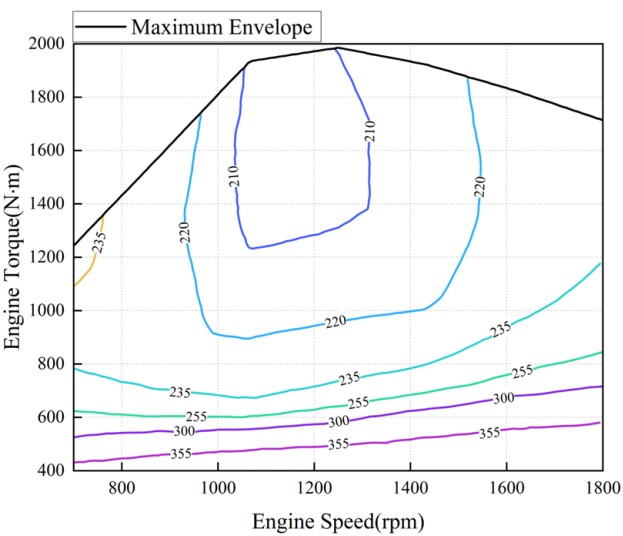

**Figure 3.** Brake specific fuel consumption map [g/kWh].

Figure 3 illustrates a BSFC map of the engine which reflects the combustion efficiency of the engine at the current moment (i.e., the mass of fuel required for every 1 kWh of effective work output). Therefore, it is necessary to optimize the energy distribution to operate the torque at the optimal efficiency point and achieve energy saving and emission reduction.

To model the relationship between fuel consumption and carbon emissions, carbon emissions need to be modeled after the modeling of fuel consumption has been completed. Modeling complex systems based on the Sigmoid function is a perfect way to do this and it is widely used in various fields. Thus, the carbon emission model will use the Sigmoid function in this paper. The carbon emissions map is shown in Figure 4.

As can be seen from [32], it has two distinct mutation points. One of them is when $N_e = 800$ rpm and the other is when $N_e = 1580$ rpm and $r_{ice} = 80\%$. At these two points, carbon emissions have a sizeable abrupt change.

Therefore, the carbon emissions $\dot{e}_c$ can be defined as:

$$\dot{e}_c = \beta_1 \left( \alpha_1 + \frac{\alpha_2}{1 + e^{-\alpha_3(N_e - 800)}} \right) r_{ice}^{\alpha_4} N_e - \beta_2 \frac{1}{1 + e^{-\alpha_5(r_{ice} - 80)}} \frac{1}{1 + e^{-\alpha_6(N_e - 1580)}} r_{ice}^{\alpha_7} N_e \tag{11}$$

where $\alpha_1, \alpha_2, \alpha_3, \alpha_4, \alpha_5, \alpha_6, \alpha_7, \beta_1$, and $\beta_2$ are fitting coefficients.

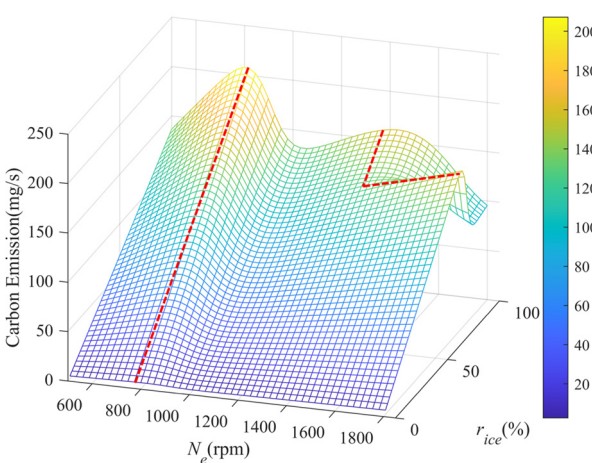

**Figure 4.** Carbon emissions map.

*2.3. Motor Output Torque Model*

The motor model can be defined by the torque distribution ratio $r_m$, as follows:

$$T_M = r_m \cdot c_m \tag{12}$$

where $c_m$ is a constant, which is a unit conversion factor.

Then, the output torque of the motor is modeled in conjunction with the angular velocity $\omega_e$ of the engine via the Willan Equation [33]

$$\begin{cases} \omega_e \cdot T_M = e \cdot P_m - P_0 \text{(Motoring)} \\ \omega_e \cdot T_M = \frac{P_m}{e} - P_0 \text{ (Generating)} \end{cases} \tag{13}$$

where $e$ and $P_0$ are the Willan constant coefficients and $P_m$ is the electric power.

*2.4. Battery Model*

The energy storage device is a ternary polymer lithium battery, the battery's nominal capacity is 45 Ah, and the total voltage is 48 V. The four lithium battery modules connected in parallel were used. The simple equivalent circuit model in reference [11] was considered as the battery model for this study. The battery is expressed as follows:

$$\frac{dSOC}{dt} = -\frac{I_B}{3600Q_B} \tag{14}$$

where $Q_B$ is the battery capacitance (unit: Ah) and $I_B$ is defined by Equation (15)

$$I_B = \frac{V_{OC} - \sqrt{V_{OC} - 4P_B R}}{2R} \tag{15}$$

where $P_B$ and $R$ are power and equivalent internal resistance of the battery, respectively; $V_{OC}$ is the open circuit voltage and it is the equation for the battery SOC as follows:

$$V_{OC} = k_1 SOC + k_2 \tag{16}$$

where $k_1$ and $k_2$ are the open circuit voltage coefficients.

*2.5. Propeller Load Torque Estimation*

The impact of irregular waves on propeller load torque is confirmed using extended Kalman filtering. In accordance with the propeller strip theory, the propeller power demand $P_P$ is proportional to the cubic engine shaft speed ($P_P = \tau N_e^3$).

By rewriting Equation (8) and defining the following observation equation based on unknown perturbation $d$, the following is obtained:

$$\dot{\omega}_e = \frac{1}{J_s}\left(T_{ICE} + T_M - \frac{d\omega_e^2}{\eta_{gb}^3}\right) \tag{17}$$

The extended Kalman filter state transition and measurement functions are as follows:

$$x[k] = f(x[k-1], u[k-1]) + w[k-1]$$
$$y[k] = h(x[k], u[k]) + v[k] \tag{18}$$

where $f$ is a nonlinear state transition function that describes how states $x = \begin{bmatrix} \omega_e \\ d \end{bmatrix}$ evolve from one time step to the next. $h$ is a measurement function relating $x$ to the measurement $y = \omega_e$ at time step $k$. $w$ and $v$ are the process noise and measurement noise, respectively. $u = \begin{bmatrix} T_{ICE} \\ T_M \end{bmatrix}$ is the set value of the control input.

By estimating the disturbance $d$, obtain $\hat{d}$. Thus, the strip theory coefficient $\hat{\tau}$ can be obtained based on $\hat{\tau} = \left(\frac{2\pi}{60}\right)^2 \frac{\hat{d}}{\eta_{gb}^3}$. The coefficient observed result is shown in Figure 5. In addition, the coefficient $\hat{\tau}$ will be used to calculate the propeller load torque, where $D_{load} = \hat{D}_{load} = \frac{60}{2\pi}\hat{\tau}N_e^2$. As can be seen Figure 5, while the strip theory coefficient fluctuates in a small range, its mean value remains almost constant and very stable. Therefore, it can be used in the proposed strategy to calculate and predict the future propeller load torque online.

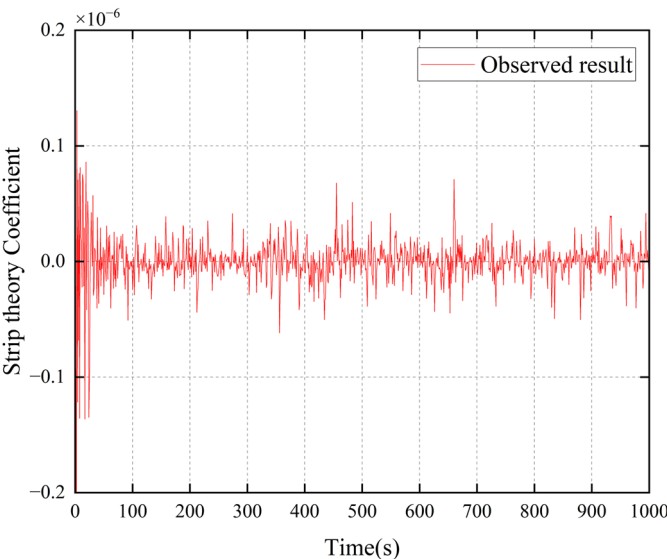

**Figure 5.** Strip theory coefficient.

## 3. Proposed NMPC Strategy via Chaotic Grey Wolf Optimization

### 3.1. Chaotic Grey Wolf Optimization (CGWO)

In this work, a chaos algorithm is introduced into GWO to overcome the trade-off between exploration and development, which is highly dependent on the global search ability of GWO. The principles of the GWO algorithm are briefly described below.

The principles of GWO are shown in Figure 6. According to Figure 6, it can be understood that grey wolves are divided into four classes. In the GWO algorithm, optimization (i.e., hunting) is guided by $\alpha, \beta,$ and $\delta$. Only $\omega$ wolves follow these three wolves. Meanwhile, $a_a, a_b,$ and $a_c$ are the three components of $\vec{a}$ in Equation (21). $C_a, C_b,$ and $C_c$ denote

the components of $\overrightarrow{C}$ in Equation (21). $D_\alpha$, $D_\beta$, and $D_\delta$ are distance vectors. Therefore, the position update of grey wolf can be expressed as follows:

$$\begin{cases} \overrightarrow{X}_a = \overrightarrow{X}_\alpha - \overrightarrow{A}_a \cdot \left| \overrightarrow{C}_a \cdot \overrightarrow{X}_\alpha - \overrightarrow{X} \right| \\ \overrightarrow{X}_b = \overrightarrow{X}_\beta - \overrightarrow{A}_b \cdot \left| \overrightarrow{C}_b \cdot \overrightarrow{X}_\beta - \overrightarrow{X} \right| \\ \overrightarrow{X}_c = \overrightarrow{X}_\delta - \overrightarrow{A}_c \cdot \left| \overrightarrow{C}_c \cdot \overrightarrow{X}_\delta - \overrightarrow{X} \right| \end{cases} \tag{19}$$

$$\overrightarrow{X}(i+1) = \frac{\overrightarrow{X}_a + \overrightarrow{X}_b + \overrightarrow{X}_c}{3} \tag{20}$$

where $i$ is the current iteration number and $\overrightarrow{X}_\alpha$, $\overrightarrow{X}_\beta$ and $\overrightarrow{X}_\delta$ are the position of $\alpha$, $\beta$, and $\delta$ wolves, respectively. $\overrightarrow{X}_a$, $\overrightarrow{X}_b$, and $\overrightarrow{X}_c$ are three position vectors. $\overrightarrow{A}_a$, $\overrightarrow{A}_b$, $\overrightarrow{A}_c$, $\overrightarrow{C}_a$, $\overrightarrow{C}_b$ and $\overrightarrow{C}_c$ are coefficients vectors and they can be calculated as follows:

$$\begin{cases} \overrightarrow{A} = 2\overrightarrow{a} \cdot \overrightarrow{r}_1 - \overrightarrow{a} \\ \overrightarrow{C} = 2\overrightarrow{r}_2 \end{cases} \tag{21}$$

where $\overrightarrow{a}$ is an encircling coefficient vector which will be linearized from 2 to 0 during the iteration of the algorithm. $\overrightarrow{r}_1$ and $\overrightarrow{r}_2$ are random vectors in [0, 1].

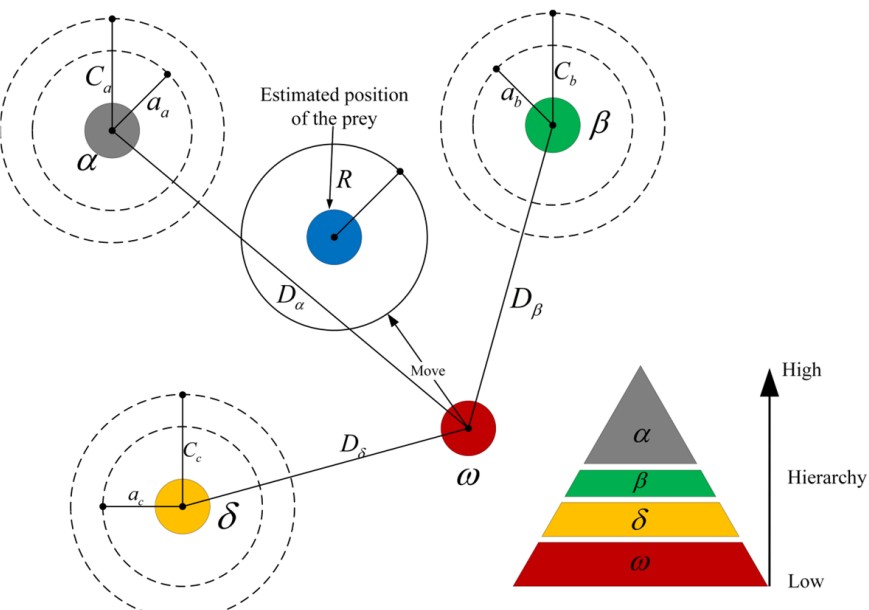

**Figure 6.** Principles of GWO.

Compared to the classical GWO algorithm, the CGWO algorithm uses a hybrid approach that combines chaotic mapping initialization and a chaotic search in order to enhance the global search capability of GWO and to increase convergence speed. According to Equation (19), the optimal solution is the average of the positions of the three leading wolves, which disregards the individual optimal solution of each wolf in this pack. Therefore, to improve the search capability, the optimal location solution for each wolf is incorporated into the search mechanism, as shown below:

$$\vec{X}_{\lambda_k}(i+1) = \vec{X}_{\lambda_k}(i) - \vec{A}\left|\vec{C} \cdot \vec{X}_{\lambda_k}(i) - \vec{X}(i)\right| \tag{22}$$

where $\lambda_k$ is the individual optimal solution of $k$-th wolf. Therefore, Equation (20) can be updated as

$$\vec{X}(i+1) = \frac{1}{3}\sum_{j=\alpha,\beta,\delta,\lambda_k}\left(1 - \frac{f\left(\vec{X}_j(i+1)\right)}{\sum_{k=\alpha,\beta,\delta,\lambda_k} f\left(\vec{X}_k(i+1)\right)}\right)\vec{X}_j(i+1) \tag{23}$$

where $f(\cdot)$ is the fitness of each wolf. The implementation steps of the algorithm are as follows:

**Step 1.** Firstly, there is a limited range in the solution space $\left[\vec{X}_{min}, \vec{X}_{max}\right]$. Then, $\vec{X}(i+1)$ needs to be mapped to range $[0, 1]$. And the map equation is followed as:

$$\Gamma_m = \frac{\vec{X}(i+1) - \vec{X}_{min}}{\vec{X}_{max} - \vec{X}_{min}} \tag{24}$$

where $\Gamma_m$ is the mapping sequence.

**Step 2.** The maximum number of iterations of chaotic maps is $\mathbb{C}_{max}$ and a set of chaotic variables is $\vartheta(n)$, $n = 1, 2, 3, \cdots, \mathbb{C}_{max}$. It is calculated by chaotic iterative mapping and the chaotic solution sequence $\vec{X}^{(n)}$ is obtained by inverse mapping.

$$\vec{X}^{(n)}(i+1) = \vec{X}_{min} + \left(\vec{X}_{max} - \vec{X}_{min}\right)\vartheta(n) \tag{25}$$

**Step 3.** According to the calculated fitness value, the optimal solution $\vec{X}^{max}$ is obtained from the chaotic solution sequence.

$$\vec{X}^{max}(i+1) = argmax\left\{f\left(\vec{X}^m(i+1)\right)\right\}(m = 0, 1, 2, \cdots, \mathbb{C}_{max}) \tag{26}$$

**Step 4.** By defining the hunting domain as $\Omega_h$, the new position update equation is:

$$\vec{X}(i+1) = \begin{cases} \vec{X}^{\mathbb{C}_{max}}(i+1) & \varepsilon \geq \Omega_h \\ \vec{X}^{max}(i+1) & \varepsilon < \Omega_h \end{cases} \tag{27}$$

where $\varepsilon$ is the random number between $[0, 1]$. The implementation flow of CGWO is shown in Figure 7. It shows the algorithms designed in this study in detail and demonstrates how the chaos algorithm is effectively combined with the GWO.

### 3.2. Control Architecture and Implementation of NMPC

NMPC solves a non-linear programming problem at each time step. A global search for the solution is performed by the CGWO algorithm, while the classical SQP algorithm is used for a further exact search to obtain the optimal control sequence. The first value of the derived control sequence is then applied to the controlled energy management system at each sampling moment. The final NMPC architecture and implementation is shown in Figure 8. The figure demonstrates all relevant aspects of this research and illustrates the application of the designed algorithms.

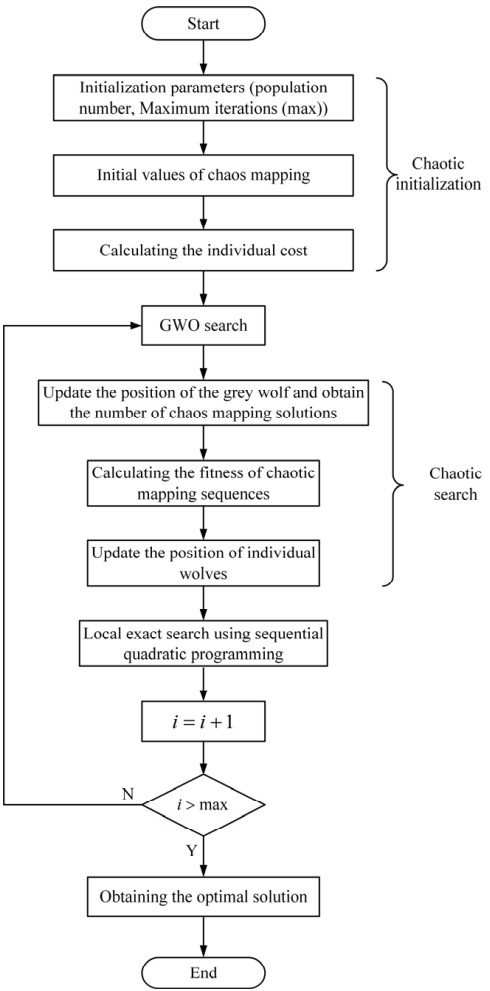

**Figure 7.** Implementation flow of CGWO.

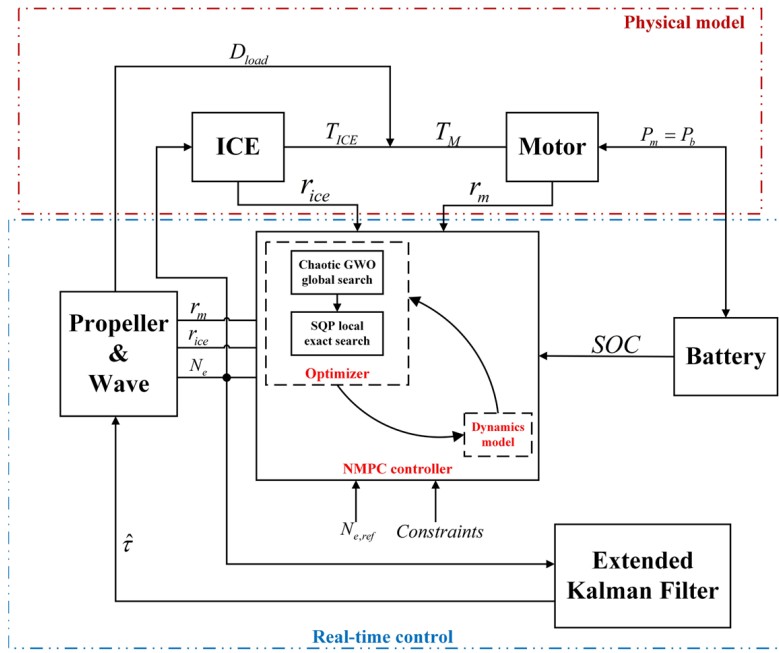

**Figure 8.** NMPC architecture and implementation.

Based on the modeling in Section 2, the dynamics model can therefore be built in a generic discrete form as follows:

$$x(k+1) = f(x(k), u(k)) \tag{28}$$

where $k$ denotes the $k$-th moment, $x = [N_e, SOC, r_{ice}, r_m]^T$, and $u = [\dot{r}_{ice}, \dot{r}_m]^T$.

In this study, the proposed NMPC is expressed as follows:

$$\begin{aligned}
\min J = & \gamma_1 \sum_{i=1}^{N-1} \left( N_{e,i} - N_{e,ref,i} \right)^2 + \gamma_2 \sum_{i=1}^{N-1} \left( SOC_i - SOC_{ref} \right)^2 \\
& + \gamma_3 \sum_{i=1}^{N-1} \dot{r}_{ice,i}^2 + \gamma_4 \sum_{i=1}^{N-1} \dot{r}_{m,i}^2 + \gamma_5 \sum_{i=1}^{N-1} r_{m,i}^2 \\
& + \varphi_1 \left( N_{e,N} - N_{e,ref,N} \right)^2 + \varphi_2 \left( SOC_{i,N} - SOC_{ref,N} \right) \\
& + \sum_{i=1}^{N} \|x_i\|
\end{aligned} \tag{29}$$

$$s.t. \quad \begin{cases}
N_{e,min} \le N_e \le N_{e,max} \\
SOC_{min} \le SOC \le SOC_{max} \\
r_{m,min} \le r_m \le r_{m,max} \\
\dot{r}_{ice,min} \le \dot{r}_{ice} \le \dot{r}_{ice,max} \\
\dot{r}_{m,min} \le \dot{r}_m \le \dot{r}_{m,max} \\
r_{ice} \ge 0
\end{cases}$$

wherein $i = 1, 2, 3, \cdots, N$ represents control point, $N_{e,ref}$ is the reference engine speed, and $SOC_{ref}$ is the reference battery SOC. $\gamma_1, \gamma_2, \gamma_3, \gamma_4, \gamma_5, \varphi_1$ and $\varphi_2$ are the penalty coefficients. In addition, $\left( N_{e,i} - N_{e,ref,i} \right)^2$ denotes the need to minimize the difference between predicted speed and reference speed to ensure the stability of the control. $\left( SOC_i - SOC_{ref} \right)^2$ is a penalty measure for deviations in the battery SOC, ensuring that energy demand changes from overcharging and discharging are avoided throughout the operating cycle. $\sum_{i=1}^{N} \|x_i\|$ is a L1 penalty term and represents all the state variables. It ensures that the GWO algorithm does not jump out of the constraint condition in the solution and can accelerate the optimization search.

## 4. Results and Analysis

In order to reduce the amount of computation, the NMPC controller sample time was set to 0.5 s. The parameters related to constraint values of the NMPC optimization problem are listed in Table 2.

All results of this experiment were obtained in the MATLAB environment. The MATLAB optimization toolbox, the open-source chaos algorithm, and the grey wolf optimization algorithm are combined and applied to implement the energy strategy as described in detail in Section 3 (the program code is described to realize the energy strategy and will be uploaded to https://github.com/NOKoooy (accessed on 19 September 2023). MATLAB is capable of solving various problems optimally and performing trade-off analysis and has good algorithm incorporation capabilities. It is also capable of performing functions such as parameter estimation, component selection, and parameter tuning, making is well-suited for energy management.

The results of the engine speed tracking and a partial enlargement are shown in Figure 9. It includes the result figures of using the classical sequential quadratic programming algorithm, the classical genetic algorithm, and the genetic algorithm combined with the SQP algorithm. As can be seen from Figure 9, all algorithms accomplish the goal of tracking the speed reference base but their fluctuation profiles differ. Firstly, the NMPC + GA algorithm has more fluctuations in track than the NMPC + SQP algorithm, the

NMPC + GA_SQP algorithm, and the designed NMPC + CGWO algorithm. RMSE is frequently used to measure prediction results in machine learning, so it can be used to measure the superiority of the tracking results of NMPC-solving algorithms.

**Table 2.** System simulation parameters.

| Description | Symbol | Value |
|---|---|---|
| Sample time | $T_s$ | 0.5 s |
| Prediction Horizon | $N_p$ | 5 steps |
| Control Horizon | $N_c$ | 5 steps |
| Penalty coefficient | $\gamma_1, \gamma_2, \gamma_3, \gamma_4,$ $\gamma_5, \varphi_1, \varphi_2$ | 1, 150, 5, 0.5 0.5, 15, 1000 |
| Battery Maximum SOC | $SOC_{max}$ | 80% |
| Battery Minimum SOC | $SOC_{min}$ | 20% |
| Maximum $N_e$ | $N_{e,max}$ | 1700 rpm |
| Minimum $N_e$ | $N_{e,min}$ | 500 rpm |
| Maximum Motor Command | $r_{m,max}$ | 90% |
| Minimum Motor Command | $r_{m,min}$ | −90% |
| Maximum Motor Command Rate | $\dot{r}_{m,max}$ | 50% |
| Minimum Motor Command Rate | $\dot{r}_{m,min}$ | −50% |
| Maximum ICE Command Rate | $\dot{r}_{ice,max}$ | 20% |
| Minimum ICE Command Rate | $\dot{r}_{ice,min}$ | −10% |
| Water density | $\rho$ | 1025 kg · m$^{-3}$ |

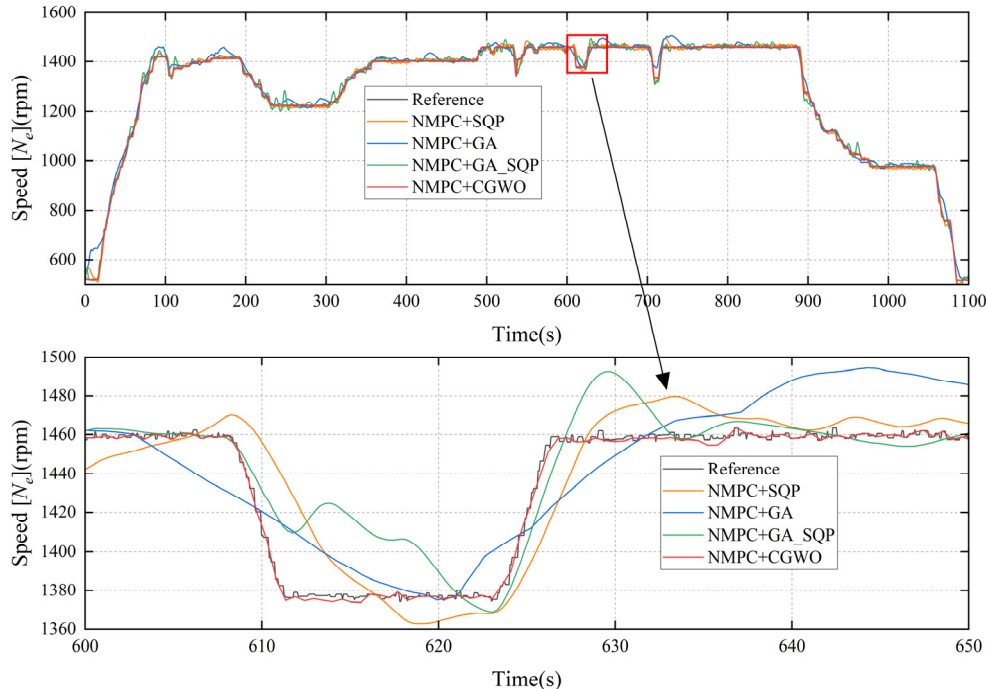

**Figure 9.** Engine speed tracking of each NMPC solving algorithm.

The RMSE values of the tracking results of the four algorithms are calculated and shown in Table 3. Combined with Figure 9 and Table 3, we can know the results of the proposed NMPC + CGWO compared with other algorithms. In contrast, it can better achieve the prediction and tracking of the engine reference speed. It can be seen that the solution effect of the GA algorithm is the most fluctuating condition and that the difference between the predicted and reference values of the model is the largest.

**Table 3.** RMSE of Solving algorithm.

| Solving Algorithm | RMSE |
|---|---|
| NMPC + SQP | 13.1597 |
| NMPC + GA | 21.9220 |
| NMPC + GA_SQP | 11.9940 |
| NMPC + CGWO | 4.5480 |

The simulated torque distribution and SOC variation of an internal combustion engine under the four algorithms are shown in Figure 10. The upper picture shows the simulated torque distribution (U) and the lower picture shows the SOC variation (L). As can be seen from the torque distribution diagram of the internal combustion engine in Figure 10, the torque provided by the proposed algorithm and GA is higher than that of the other two algorithms under the same working conditions. The strategy designed by the institute provides higher torque than GA before 500 s. It shows the advantages of the designed algorithm. While the torque changes, so does the battery SOC. The battery SOC variation figure shows that the GA algorithm solves for the optimized battery SOC fluctuation range of 49.4–52.9% and that of the classical SQP algorithm is 47.9–52.7%. The GA-SQP algorithm solves for the battery SOC fluctuation range of 44.8–50.6% and the fluctuation range of battery SOC based on chaotic grey wolf NMPC strategy is 46.8–53.3%. The less fluctuation in battery SOC is owing to the entire ship operating conditions being not fully considered. The SOC needs to be stabilized to near the reference value each time step, which limits the variation of battery SOC.

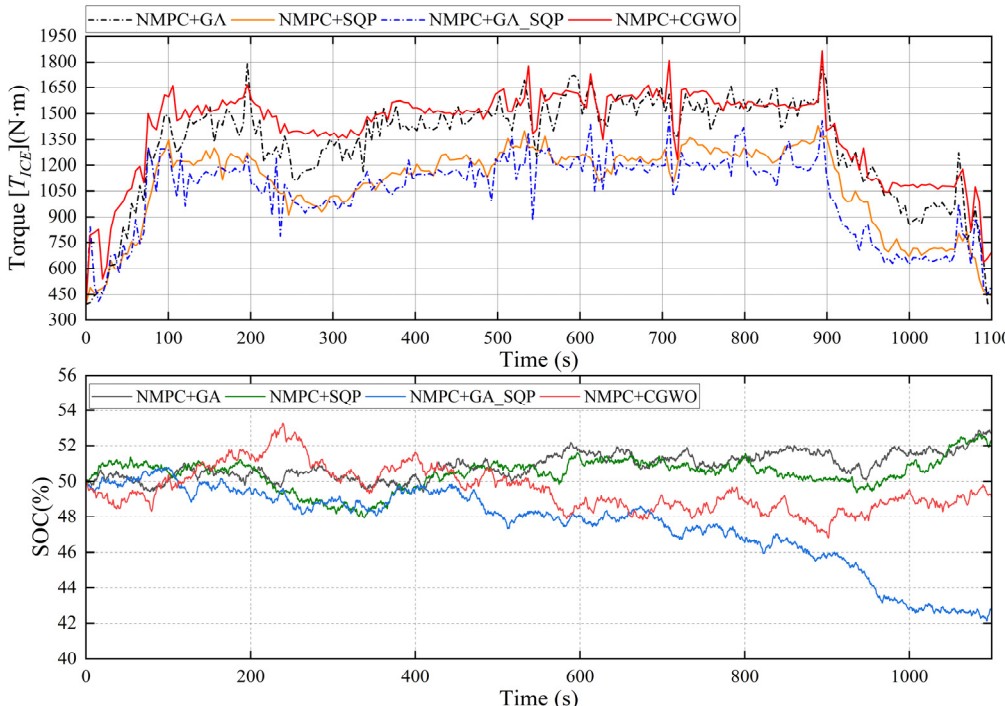

**Figure 10.** (U)Simulate ICE torque distribution (N·m) and (L)battery SOC variation (%).

The ultimate goal of designing energy management strategies is lower fuel consumption and reduced emissions. Therefore, comparing fuel consumption and carbon emissions must be performed using the developed NMPC optimization solution algorithm. Fuel consumption curves (U) and carbon emission curves (L) for all methods are shown in Figure 11.

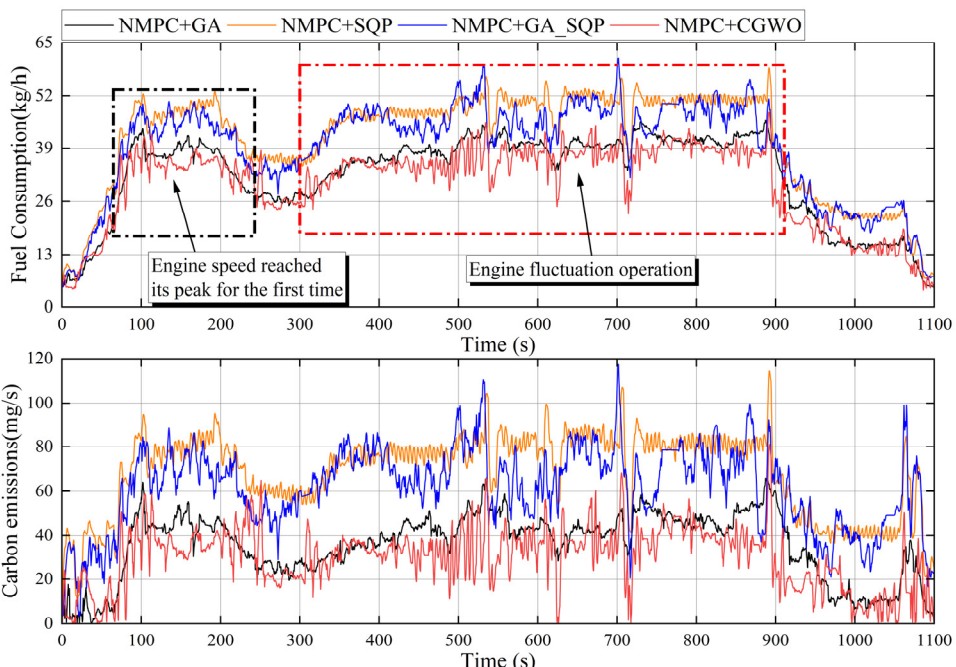

**Figure 11.** (U)Fuel consumption (kg/h) and (L)carbon emissions (mg/s).

According to Figure 11, the fuel consumption curve and carbon emission curve of NMPC + CGWO are below the corresponding fuel consumption and carbon emissions of other methods. Two nodes are labeled in Figure 11 and, based on the engine speed tracking operating conditions shown in Figure 9, it is known that the first peak in fuel consumption is reached around 60 s to 230 s for all four methods. After 300 s, it enters a phase similar to a plateau period according to the set-up operating conditions.

However, both the optimized fuel consumption curve and the carbon emission curve are accompanied by large fluctuations. This is because the engine speed itself is made up of many small gradient changes. In Section 2, $\dot{m}_f$ and $\dot{e}_c$ are differential forms. Therefore, when calculating fuel consumption and carbon emissions, it is necessary to calculate the integral based on the optimized operating conditions, this leads to large fluctuations. Although there are fluctuations, it still demonstrates the superiority of the algorithm designed in this study.

In order to compare the performance of the four algorithms more intuitively and accurately, the total fuel consumption, total carbon emissions, and total computation time of the four algorithms during the optimization time period are presented in Table 4, where the comparison of computation time demonstrates the real-time performance of the algorithms.

**Table 4.** Total fuel consumption, total carbon emissions, and total computation time.

| Algorithm | Total Fuel Consumption (kg) | Total Carbon Emissions (g) | Total Computation Time (s) |
|---|---|---|---|
| NMPC + GA | 9.8779 | 39.1280 | 344.7736 |
| NMPC + SQP | 12.5982 | 75.6191 | 26.7451 |
| NMPC + GA_SQP | 11.9610 | 67.7434 | 104.3345 |
| NMPC + CGWO | 9.3184 | 33.0421 | 38.25141 |

As shown in Table 4, the proposed strategy corresponds to the lowest total fuel consumption and carbon emissions, indicating that its solution is the best. Although it is accompanied by large fluctuations, as shown in Figure 11, it does not impact the NMPC energy management strategy much. Among the other three algorithms, the one with the best fuel consumption and carbon emissions performance is the GA algorithm. In addition,

the total computing time of the proposed NMPC strategy ranks second among the four algorithms, only higher than that of the SQP algorithm. But the fuel consumption and carbon emissions of the SQP solving algorithm are higher than those of the proposed solving algorithm. To further illustrate the effectiveness of the strategy, the engine load paths under various algorithms are shown in Figure 12.

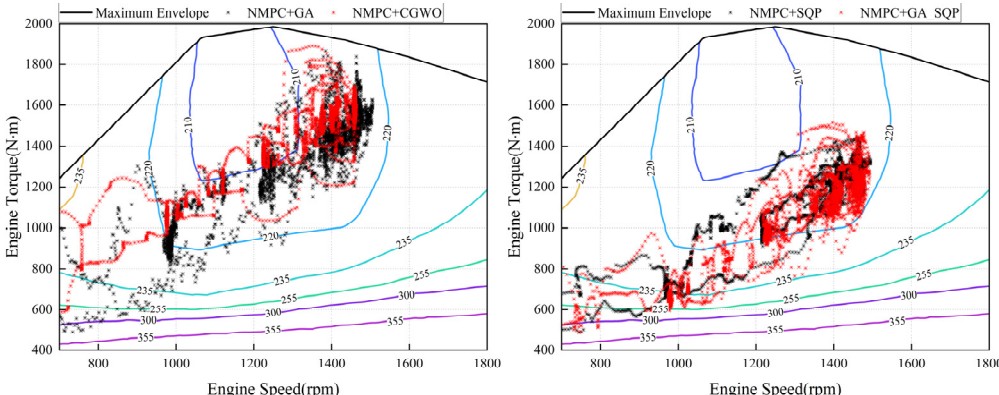

**Figure 12.** Engine load paths under four algorithms.

According to Figure 12, it is known that the engines under the two solution algorithms, SQP and GA-SQP, did not approach the operating point with a lower effective fuel consumption rate as in the case of GA and the proposed algorithm. The proposed strategy is more effective than GA. It also corroborates the argument that the proposed strategy in Table 4 has the lowest fuel consumption and, moreover, reflects the effectiveness of the proposed strategy.

## 5. Conclusions

This study is based on previous research [11] and designs a nonlinear model predictive control energy management strategy using a hybrid optimization algorithm that combines a chaos algorithm with GWO. The hybrid optimization algorithm is utilized for global search, while the classical SQP is employed for local exact search. Furthermore, an L1 penalty factor is introduced to prevent the global search from crossing the boundaries. This strategy ensures optimality to the maximum extent in order to achieve a reasonable distribution of energy. In addition, irregular waves are modeled and extended Kalman filtering is used to estimate ship propeller load torque as part of a nonlinear prediction model.

In addition, the results show that the proposed NMPC based on the chaotic grey wolf optimization algorithm yields better performance compared to other algorithms. Compared to using the classical SQP strategy, the proposed algorithm in this study reduces fuel consumption by approximately 26% and carbon emissions by approximately 56% over the optimized time period. Despite a slight increase in the total optimization time, it exhibits better performance in the prediction and tracking of engine speed (with a RMSE value of 4.5840), indicating good immunity to interference. Therefore, the proposed NMPC strategy can rationalize the energy allocation and provide ideas for realistic applications.

The future research work will focus on the use of artificial neural network methods to predict the energy consumption of ships with respect to the actual hardware implementation of energy management strategies.

**Author Contributions:** Conceptualization, L.C. and D.G.; methodology, L.C. and D.G.; software, L.C.; validation, L.C.; investigation, D.G. and Q.X.; data curation, L.C. and D.G.; writing—original draft preparation, L.C.; writing—review and editing, D.G. and Q.X.; visualization, L.C.; supervision, D.G. and Q.X. All authors have read and agreed to the published version of the manuscript.

**Funding:** This research was funded by the Shanghai Science and Technology Project of China (Grant No. 20040501200, 21DZ205803) and the National Natural Science Foundation of China (NSFC) (Grant No. 61673260 and 61304186).

**Institutional Review Board Statement:** Not applicable.

**Informed Consent Statement:** Not applicable.

**Data Availability Statement:** The paper codes and data will be uploaded to https://github.com/NOKoooy (accessed on 19 September 2023).

**Acknowledgments:** The authors thank the anonymous reviewers for suggesting valuable improvements to the paper.

**Conflicts of Interest:** The authors declare no conflict of interest.

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
