# Peer review of "Energy Management Strategy of Hybrid Ships Using Nonlinear Model Predictive Control via a Chaotic Grey Wolf Optimization Algorithm"

_jmse, doi:10.3390/jmse11091834_

Round 1
Reviewer 1 Report
The paper is fairly understandable, although it has some grammatical errors. I am not sure about the significance of its novelty but it can be considered a novelty.
Based on my opinion, some of the notations need to be defined better.
Line 133: "As shown in Figure 2, a parallel hybrid ship was applied in this study." Figure 1 or Figure 2? If figure 2 should remain, my advice is to swap figures 1 and 2 to improve clarity. It's a bit confusing like this.
Line 170: The sum of wave counts is as follows: .... (L is the wave-170 length). Is it correct? I regret to say that I can’t figure out what the sum of wave counts is and how it is connected with this equation.
Line 156: By estimating the disturbance d , obtain ˆd . Please provide some more details on what it is ˆd .
Line 266: "The principles of the GWO algorithm are briefly described below."
I don't agree with this sentence since in the following of the paper there are a lot of equations and parameters that were not described adequately. Young readers can follow it very hard.
Line 320: represents the cost function. With this function and penalty coefficients, the authors transformed a multi-objective optimization problem into a single-objective problem.
Is it possible to add these two goals since each of them represents a different physical quantity (have different units of measurement)?
How did you choose the penalty factors?
Line 321: "... a L1 penalty term and represents all the state variables." - It is not clear what state variables are. What does xi represent from the following notation?
In general, the work should be improved and attention should be paid to the correctness, grammar and structure of the work.
Reviewer 2 Report
The author's work is part of a project in which they participate and contributes to topics that lead to the reduction of harmful gas emissions and energy efficiency. The development of hybrid technologies, which offers a new direction for the rational distribution of energy, led to the development and use of different methods in the management of energy resources on board. So, this paper describes the management of Hybrid ships using Nonlinear Model Predictive Control.
The work presented by the authors largely coincides with the work of Long Chen, Diju Gao, Qimeng Xue. "Energy management strategy for hybrid power ships based on nonlinear model predictive control", International Journal of Electrical Power & Energy Systems, 2023.
The authors must somehow determine how to present a part of the work that is already available on the page:
https://www.sciencedirect.com/science/article/abs/pii/S0142061523003769?via%3Dihub#preview-section-snippets
It is necessary to leave out a part of the text that is repeated and refer to the literature that is already available (mentioned article).
In general, the authors must once again check all equations and explanations of individual parameters because some are missing and some labels are not fully indicated (equation and explanation).
The authors did not indicate from which literature source they took the equations, e.g. 1, 4...whether from Thesis for the degree of philosophise doctor (Trondheim, April 2008 Norwegian University of Science and Technology, Faculty of Information Technology, Mathematics, and ElectricalEngineering, Department of Engineering Cybernetics) Luca Pivano: Thrust Estimation and Control of Marine Propellers in FourQuadrant Operations or some other source. It is necessary to indicate the source of the literature before writing it down.
The authors do not use the same symbols, for example in Table 1 is the symbol for added mass
ms and in the equation 8 is mo is the added ship's mass. It is necessary to equalize everything together.
Also, the pictures are not the whole explanation, the authors only refer to them, for example, picture 5 or 6. A short explanation of the picture is needed, not only for these larger and other pictures.
It is necessary to follow up on this previous research and more clearly emphasize the difference, that is, what is new compared to the previous work. This work is a continuation of the research, and the research will certainly go further, so maybe the authors should write in the conclusion what they will do next and not repeat part of the discussion.
It is necessary to supplement the list of reference literature.
Authors, please check the text, sentences...
Reviewer 3 Report
The article of the authors is devoted to the issue aimed at developing a strategy for managing the energy consumption of hybrid ships using modern software.
The shipping industry makes a tangibly significant contribution to the total carbon dioxide emissions into the global air basin due to the combustion of fossil hydrocarbon fuels by ship power plants. In April 2018, the International Maritime Organization adopted a new strategy to reduce greenhouse gas emissions for shipping, which aims to phase in average carbon dioxide emission intensity per ton-mile by 40% by 2030 and by 70% by 2050.
In this regard, it is necessary to consider the main directions for improving the modern fleet, including improving the logistics support for transportation, further improving the quality of the ship's hydrodynamic parameters, improving the characteristics of power generating equipment and units of ship power plant systems, replacing used fossil fuels with alternative ones that are potentially capable with an integrated approach. ensure the fulfillment of the tasks set by the strategy. It is necessary to pay attention to the fact that the use of alternative fuels is of particular importance as a direction that can provide up to 100% reduction in greenhouse gas emissions. Various alternative fuel options should be considered, including liquefied natural gas; carbon neutral fuels based on carbon: biofuels and synthetic fuels (electric fuels); carbon-free fuels, as well as to analyze the prospects for their use.
The studies presented in the paper are undoubtedly of interest to readers in the field under consideration.
However, it would be necessary to clarify a number of comments that are available to the article:
1. In the introduction, the literature review should be expanded by considering methods for predicting power consumption based on machine learning methods, which is especially important for maritime transport due to the lack of many initial data and their random nature. In particular, it was possible to analyze the use of artificial neural networks, as can be seen from the following works: https://doi.org/10.1016/j.apor.2021.102964, https://doi.org/10.21177/1998-4502-2022-14-3-486-493, https://doi.org/10.3390/en15238919.
2. The article should include the section "Materials and Methods", in which it is necessary to more clearly identify the main methods used to conduct complex experimental studies.
3. It would be possible to cite the numerical values of carbon dioxide emissions into the atmosphere as a result of water transport in various regions of the world and conduct an appropriate comparative analysis, where it is the research to reduce emissions that is most relevant and significant.
4. In the section “2. Hybrid power system description and modeling” should compare the studied parameters of the ship (Table 1) with other water transport and the possibility of conducting similar studies on other ships. Is a wider variation of the parameters presented in table 1 allowed?
5. A more detailed analysis of Figure 5 should be provided, which shows a brake specific fuel consumption map.
6. Is a patent expected for the scheme shown in Figure 8?
7. The article should explain how the optimization problem was solved using the MATLAB software. It should show the advantages of using this program in comparison with similar software tools and the feasibility of using it to analyze ship parameters (page 12).
8. Have you considered the possibility of predicting the ship's energy consumption using artificial neural networks?
9. According to the dependencies presented in Figures 9, 10, 11, mathematical models of the considered output parameters, determination coefficients and predictive values should be given.
10. Conclusions should include specific numerical results obtained in the work and dwell on further prospects for conducting research in this area.
Round 2
Reviewer 1 Report
I want to thank the authors for carefully considering my comments.
I have no more comments and I believe the manuscript can be accepted for publishing.
Author Response
Thank you very much for your positive opinion about of our work, and sincerely thank you for your time and careful review.
Reviewer 2 Report
Thanks to the authors for the corrections they made.
They corrected part of the text they downloaded from https://www.sciencedirect.com/science/article/abs/pii/S0142061523003769 (Journal of Electrical Power & Energy Systems, 2023; Long Chen's, Diju Gao, Qimeng Xue: "Energy management strategy for hybrid power ships based on nonlinear model predictive control") . Both works represent work on a project, they are interesting, but they should have simply indicated the differences in individual works. Namely, they could simply write briefly what they did in this work and what is different in the work they presented in this part of the JMSE .
The authors have made appropriate corrections.
It is recommended that they check the entire text once more.
Reviewer 3 Report
The necessary changes have been made. I recommend the article for publication.
Author Response

(The authors gave the same response as above.)
